# Feeding Malic Acid to Chickens at Slaughter Age Improves Microbial Safety with Regard to *Campylobacter*

**DOI:** 10.3390/ani11071999

**Published:** 2021-07-05

**Authors:** Fangzhe Ren, Wenbin Yang, Juanjuan Hu, Pingyu Huang, Xin-An Jiao, Jinlin Huang

**Affiliations:** 1Jiangsu Key Lab of Zoonosis/Jiangsu Co-Innovation Center for Prevention and Control of Important Animal Infectious Diseases and Zoonoses, Yangzhou University, Wenhui East Road 48, Yangzhou 225009, China; fzren@yzu.edu.cn (F.R.); jiao@yzu.edu.cn (X.-A.J.); 2Key Laboratory of Prevention and Control of Biological Hazard Factors (Animal Origin) for Agri-Food Safety and Quality, Ministry of Agriculture of China, Yangzhou University, Wenhui East Road 48, Yangzhou 225009, China; DZ120200009@yzu.edu.cn (W.Y.); MZ120191184@yzu.edu.cn (J.H.); MX120200895@yzu.edu.cn (P.H.); 3Joint International Research Laboratory of Agriculture and Agri-Product Safety, Yangzhou University, Wenhui East Road 48, Yangzhou 225009, China

**Keywords:** poultry production, malic acid, *Campylobacter*, microbial safety

## Abstract

**Simple Summary:**

Chicken meat has become a popular food that is consumed worldwide. However, chicken flocks suffer from *Campylobacter* infection during their rearing period. *Campylobacter* is the most serious pathogen colonizing chicken flocks which could be transmitted through the food chain and threaten public health. The traditional strategy of using antibiotics to inhibit pathogens in chicken flocks is no longer acceptable due to the increasing risk of antibiotic resistance. Thus, finding alternative antimicrobial agents has become a priority in recent years. In this study, malic acid was supplied to flocks in order to find an effective means of reducing the contamination of *Campylobacter* and to evaluate its potential effects on poultry production. By using malic acid-supplemented drinking water for 5 days before slaughtering, the *Campylobacter* carriage was significantly decreased in the treated group compared to the control group. Malic acid has no adverse effects on chickens, though it could change the composition of chicken meat by increasing the moisture content and decreasing the fat content and it could be applied as a potential antimicrobial agent in poultry production.

**Abstract:**

This study supplied malic acid-supplemented drinking water to flocks that were naturally *Campylobacter*-positive and assessed the effect of feeding malic acid to chickens on *Campylobacter* reduction and poultry production. In Experiment 1, chickens were provided with malic acid-supplemented drinking water for three weeks. The contamination loads of *Campylobacter* were decreased by 0.91–0.98 log after the first week of use (*p* < 0.05). However, this effect did not persist over time and significant decontamination could not be found in the second and third weeks of application. Thus, in Experiment 2 malic acid-supplemented drinking water was given to chickens for a period of five days at slaughter age. The *Campylobacter* carriage was found to be effectively decreased by 1.05–1.55 log (*p* < 0.05). Malic acid had no adverse effects on chicken body weight, weight gain, intestinal indices, or the microbiota. In addition, it could change the composition of chicken meat since the moisture content was increased by 5.12–5.92% (*p* < 0.05) and the fat content was decreased by 1.60% (*p* < 0.05). Our study provides an effective means for reducing the contamination of *Campylobacter* during the chicken rearing period and this method can be applied to promote the safe development of poultry farming and its products.

## 1. Introduction

Poultry production is closely related to human life and chicken meat has become a popular food that is consumed worldwide due to its high protein content, ample levels of micronutrients, low fat content, and relatively low price when compared to beef or pork [1,2]. However, during the rearing period, chicken flocks suffer from foodborne pathogen infections, which could be transmitted through the farm-to-fork food chain and threaten food safety and public health. This issue has been one of the major problems that has beset chicken meat production, with chickens’ contamination with *Campylobacter* being particularly serious [2,3,4].

The chicken is generally recognized as the natural host for *Campylobacter* and does not develop symptoms [5]. During the slaughter process, the chicken’s intestinal tract may leak or rupture and contaminate the meat by cross-contamination [2,3,4]. It is estimated that handling poultry and consuming contaminated chicken meat products may account for 20% to 30% of human infection cases, while 50% to 80% may be attributed to the chicken reservoir as a whole [6]. Recent studies have shown that the decontamination of *Campylobacter* during the rearing period would significantly reduce the risk of human campylobacteriosis [4].

Due to the increase in antibiotic resistance in pathogens, the use of antibiotics in chicken production has been restricted [7,8]. Thus, finding alternative antimicrobial agents has become more and more important in recent years [3]. Malic acid is a dicarboxylic organic acid which has antimicrobial activity [9]. Moreover, malic acid is an intermediate in the metabolic cycles of organisms for energy production and could aid digestion and absorption by chelating various cations and enhancing the activities of digestive enzymes [10]. Malic acid is deemed as GRAS (Generally Recognized As Safe), which means that it does not have adverse effects on human intake or animal feeding when properly used. The application of malic acid has also been shown to have its potential benefits in promoting the performance of animals and improving the quality of products derived from them [10,11,12,13].

The bactericidal effect of malic acid on *Campylobacter* has been demonstrated in laboratory culture and raw meat samples [14,15]. However, its antimicrobial efficacy against *Campylobacter* during the rearing period of chickens has never been investigated and the potential effects of supplementing chicken drinking water with malic acid on the performance of chickens and the quality of their products also needs to be assessed. This study added malic acid to the drinking water of chicken flocks that were found to be naturally *Campylobacter*-positive. The effect of malic acid was evaluated during the process of poultry production to provide a reference for the further practical application of malic acid in order to promote the safe development of poultry farming and to improve the quality of poultry products.

## 2. Materials and Methods

### 2.1. Animals and Treatments

Arbor Acres (AA) broilers were provided by Jiangsu Jinghai Poultry Industry Group Co., Ltd. (Nantong, Jiangsu, China), and they were raised in floor pens with dimensions of approximately 8 m^2^ per group. Partridge chickens were provided by poultry farms in Nantong city, Jiangsu province, and raised in cages with dimensions of approximately 3.5 × 1.75 m per group. For all chicken houses, the temperature was maintained at around 21–28 °C and the humidity at around 50–60%. The flocks were supplied with commercial feed with corn, wheat, and soybean meal as the main ingredients. The detailed composition of the feed is listed in Appendix A. Normal water or acidified water prepared by supplementation with malic acid with a final pH value of approximately 4.0 was provided to flocks in different groups for drinking. Chickens had free access to water and commercial feed at all times and the feed and drinking water were refreshed every day. All the experimental and animal management procedures were approved by the Animal Welfare and Ethics Committees of Yangzhou University and complied with the guidelines of the institutional administrative committee and ethics committee of laboratory animals (SYXK [Su] 2016–0020).

### 2.2. Experimental Design

Two experiments were carried out in this study and the experiments were performed where the chickens were raised. In each experiment, chickens with few individual differences and of both sexes were randomly divided into two groups and reared in separate areas with the same environmental conditions. The experimental group was provided with malic acid-supplemented water (pH 4.0), while the control group was given non-supplemented water. The source and concentration of the malic acid used in each experiment was the same. The experimental procedure and data analysis were performed utilizing a blind method, which means that the investigators did not know the group information in advance. Cloacal swabs were sampled for each flock before conducting the experiments to ensure that the chickens were *Campylobacter*-positive.

Experiment 1. Two flocks (2 week old AA broilers and partridge chickens) were selected for use in the evaluation of the antimicrobial effect of malic acid-supplemented drinking water on *Campylobacter* during the chicken rearing period. In each flock, 40 chickens were divided into 2 groups (20 chickens per group), then the acidified water and non-supplemented water were given to the experimental group and control group, respectively, and continuously supplied for three weeks. The cloacal swab samples of the chickens were collected before conducting the experiment and after one, two, and three weeks of being supplied with supplemented or non-supplemented water, they were used to determine the contamination load of *Campylobacter*.

Experiment 2. Another two flocks (5 week old AA broilers and 10 week old partridge chickens) were selected for use in the evaluation of the application of the acidified water at slaughter age. In each flock, 40 chickens were divided into 2 groups (20 chickens per group) and the experimental group and control group were given the acidified water and non-supplemented water, respectively, for five days. The body weights and cloacal swab samples of each chicken were recorded before conducting the experiment and five days after the experiment. The AA broilers were sent to a slaughter house after the experimental period. The broilers were electrically stunned and killed by neck cutting and after the defeathering process, intestine and meat samples were collected and analyzed.

### 2.3. Enumeration of Campylobacter and Microbiota

The number of *Campylobacter* and the number of microbiota present in the feces were calculated as described previously with some modifications [16]. The cloacal swabs were moistened in buffered peptone water (BPW) and weighed both before and after taking the fecal samples, then placed in a Cary–Blair transport medium and transported to the laboratory. The cloacal swab with the sample was immersed in 1 mL of phosphate buffer saline (PBS) for 20 min and shaken several times. The swab was then removed from the solution, appropriately diluted with 100 μL of the diluent; spread onto the *Campylobacter* blood-free containing charcoal cefoperazone deoxycholate agar (CCDA) containing rifampicin, polymyxin B, trimethoprim, cycloheximide, vancomycin, and amphotericin B; and then incubated at 42 °C under a microaerophilic atmosphere for 36 h to count the *Campylobacter*. For the enumeration of the intestinal microbiota of the chicken, 100 μL of the appropriate diluent prepared as described above was spread on the plate count agar (PCA) and the plate was incubated at 37 °C for 24 h to conduct a colony count. The determination of the *Campylobacter* colonization in the caecum was performed as described in a previous study [17]. The caecum samples of the chicken were collected during the slaughter (experiment 2) and the lumenal contents were gently extruded. The tissue was weighed, homogenized, serially diluted, and plated on selective CCDA agar to count the *Campylobacter* as described above.

### 2.4. Analysis of the Chicken Performance

The body weights of the chickens were recorded before and after giving the chickens the acidified water to analyze its influence on the chickens’ weight gain. During the slaughter of AA broilers (experiment 2), the intestines of the chickens were collected, the total weights of the intestine samples were recorded, then they were separated and ligated at the subsections to compare the lengths of the intestinal components. The caecal contents were squeezed into 10 mL tubes, diluted (1:8) with distilled water, vortexed, and read using a pH meter as described previously [18].

### 2.5. Analysis of the Proximate Composition of Chicken Meat

The chicken meat samples were collected during slaughter (experiment 2). Approximately 10 g of breast and thigh meat samples were collected and stored at −70 °C. The samples were thawed overnight at 4 °C and ground before being used for subsequent analysis. The moisture present in the samples was determined by drying the meat samples at 105 °C until a constant weight was achieved. Crude proteins were determined according to the Kjeldahl method from the amount of ammonia ions neutralized by sodium hydroxide. Crude fat was determined by the Soxhlet extraction procedure with petroleum ether. Crude ash was determined by burning the samples in a muffle furnace at 550 °C for 12 h. All procedures were performed according to the method described by the Association of Official Agricultural Chemists (AOAC) [19].

### 2.6. Statistical Analysis

Statistical analyses were performed by using the GraphPad Prism software 8.0 (San Diego, CA, USA). Data were expressed as the mean ± standard deviation (SD). Normality was tested with the Kolmogorov–Smirnov test or with the D’Agostino–Pearson test to assure the Gaussian distribution of values. The statistical significance between the two groups was analyzed using the unpaired *t*-test (Welch’s correction was used if there were unequal variances) when the normality test passed, otherwise the Mann–Whitney U test was used and 95% confidence intervals were applied. *p* < 0.05 was considered statistically significant.

## 3. Results

### 3.1. The Reduction Effect of Malic Acid-Supplemented Drinking Water on Campylobacter Was Significant in the First Week of Use and Decreased with Extended of Time Supplementary

In Experiment 1, malic acid-supplemented drinking water was provided to flocks (2 week old) continuously for three weeks. Before the experiment, the *Campylobacter* carriages in the control group and the malic acid-treated group were similar in both the partridge chickens and AA broilers (*p* > 0.05). After one week of treatment with acidified drinking water, the *Campylobacter* carriages were found to be significantly decreased by 0.98 log and 0.91 log in the partridge chickens and AA broilers, respectively, than compared to the control group (*p* < 0.05). After two weeks of malic acid treatment, the average amounts of *Campylobacter* in the acid-treated groups were slightly higher than in the control group for both the partridge chickens and AA broilers, but no significant differences were observed (*p* > 0.05). Similar results were also observed in the third week, suggesting that the decontamination effect of the malic acid-supplemented water did not persist over time but rather decreased over more extended periods of use (Figure 1).

### 3.2. The Use of Malic Acid-Supplemented Drinking Water for Five Days before Slaughter Is a Feasible Method to Reduce the Contamination of Campylobacter in Flocks

For Experiment 2, the acidified drinking water was given to flocks at the slaughter age over five days (5 week old AA broilers and 10 week old partridge chicken). Before the experiment, similar carriages of *Campylobacter* of approximately 4 log_10_ CFU (Colony Forming Units)/g (gram) were detected in all groups (*p* > 0.05). After five days of treatment, for the partridge chicken the *Campylobacter* carriage detected in the cloaca of the control group was 3.53 log_10_ CFU/g, which was similar to the data before the experiment. However, in the malic acid-treated group, the *Campylobacter* carriage was significantly decreased to 1.98 log_10_ CFU/g and a 1.55 log reduction was found when compared to the control group (*p* < 0.05) (Figure 2A). In the AA broilers, the *Campylobacter* carriage in the cloaca of the control group was increased to 5.19 log_10_ CFU/g during the five days of the experimental period, while the carriage of the malic acid-treated group was decreased to 4.14 log_10_ CFU/g and a 1.05 log reduction was found (*p* < 0.05) (Figure 2B). In caeca samples of AA broilers, the *Campylobacter* colonization load of the control group was 9.96 log_10_ CFU/g while the corresponding colonization level in the malic acid-treated group was only 8.40 log_10_ CFU/g, showing a 1.56 log reduction (*p* < 0.05) (Figure 2C).

### 3.3. The Treatment of Malic Acid-Supplemented Water Does Not Influence the Chicken Performance, Intestinal Indices, and Microbiota

In Experiment 2, the average weight and microbiota of flocks in different groups were similar before the experiment (*p* > 0.05). For the partridge chickens, the average weight of the malic acid-treated group was approximately 813.3 g and the weight gain was approximately 25.3 g/day. These values are slightly higher than those of the control group after being treating with the malic acid-supplemented water for five days (*p* > 0.05). In AA broilers, the average weights of the malic acid-treated group and the control group were approximately 1523.8 and 1504.1 g, respectively (*p* > 0.05), and the weight gains were approximately 72.6 and 76.6 g/day, respectively (*p* > 0.05) (Table 1). For the intestinal microbiota, the values in the cloaca of partridge chickens were slightly decreased to 11.66 log_10_ CFU/g in the malic acid group after five days of treatment and 11.72 log_10_ CFU/g in the control group (*p* > 0.05). In AA broilers, the detected microbiota in the malic acid treated group and control group were 11.79 and 11.51 log_10_ CFU/g, respectively (*p* > 0.05), suggesting that malic acid does not affect the body weight and microbiota of flocks (Table 1).

The potential effect of malic acid-supplemented water on the intestines of chickens was evaluated in AA broilers. As shown in Table 2, after the 5 day experimental period the average weight of the intestine was approximately 25.1 g, the length of the small intestine was approximately 112.6 cm, the length of the caecum was approximately 11.0 cm, and the pH of the caecal content was approximately 7.26 in the control group. Meanwhile, the weight of the intestines in the malic acid-treated group was approximately 26.7 g, the length of the small intestine was approximately 110.9 cm, the length of the caecum was approximately 12.1 cm, and the pH of the caecal content was approximately 7.03. No significant differences were observed between the treatment group and the control group (*p* > 0.05).

### 3.4. Drinking Malic Acid-Supplemented Water Changes the Composition of Chicken Meat

In Experiment 2, the meat composition of AA broilers was evaluated by measuring the contents of moisture, crude protein, ash, and fat. As shown in Table 3, there were no significant differences in the protein and ash contents between the group supplemented with malic acid and the control (*p* > 0.05). However, compared to the control group, the moisture content in the malic acid-treated group increased by 5.12% in the thigh meat portion and 5.92% in the breast meat portion, respectively (*p* < 0.05). The fat content in the malic acid-treated group decreased by 1.60% for the thigh meat compared to the control group (*p* < 0.05).

## 4. Discussion

*Campylobacter* is one of the leading causes of food-borne gastroenteritis in humans worldwide [2,3,4], which accounts for approximately 96 million cases of human illness per year on a global scale [20]. Poultry is the most common species associated with human *Campylobacter* illness. Most chicken flocks became *Campylobacter*-positive at slaughter age, rendering them an important reservoir for human infection. After *Campylobacter* infects humans, the clinical symptoms can include mild abdominal pain, headaches, fever, vomiting, and severe watery and bloody diarrhea. The infection can sometimes result in serious sequelae, such as Guillain–Barré syndrome, Miller Fisher syndrome, and reactive arthritis [20]. Although most cases are self-limiting, a number of patients will require medication and hospitalization, thus representing a great health and economic burden for the public [3,4,20]. Therefore, it is important to control *Campylobacter* contamination at the farm level.

Malic acid can be industrially produced and has the advantages of causing no pollution/residue, not being toxic, and having an easy application [21]. In previous studies, it was found that malic acid could cause a 6 log reduction in *Campylobacter* in laboratory broth and a 4 log reduction in chicken juice after 24 h of exposure at 4 °C [14]. The contamination of chicken legs with *Campylobacter* was also observed to decrease 1.18 log after treatment with malic acid solution at 4 °C for 8 days [15]. These results indicate the potential of the application of malic acid in the poultry industry. Malic acid is commonly recognized as a mild acid and has been widely applied in the food industry [21], which also makes it possible to apply in animal feeding. Our pre-experiment found that the effect of malic acid against the growth of *Campylobacter* was obvious in vitro (Appendix A). Moreover, an in vivo study showed that malic acid had a more stable effect on controlling *Campylobacter* contamination than compared to other acids (Appendix A). The minimum inhibitory concentration of malic acid against *Campylobacter* was also found to be lower than for other acids [22], which suggests that malic acid could be more effective for treating *Campylobacter* contamination in poultry production.

Organic acids exploit their antimicrobial activity in the undissociated form, which is closely related to the pH of the medium [23]. Previous studies have indicated that organic acids could show bactericidal effects on *Campylobacter* strains at a pH of 4.0 [24], thus we decided to adjust the pH to 4.0 using malic acid in this study. There is no evidence of the vertical transmission of *Campylobacter* in chicken flocks, while cross contamination from the environment has been regarded as an important infection source [4,25]. Malic acid could be added to the feed or drinking water of broilers. The dry condition of feed is lethal to *Campylobacter*, which is not regarded as a potential source for contamination [26]. Meanwhile, water is an important vehicle for spreading *Campylobacter* and is prominent in chicken flocks [4,26,27]. Thus, malic acid was added to drinking water in our study.

Although most studies have reported the effectiveness of using organic acid to control the contamination of pathogens during animal rearing [18,22,27,28,29], some results have also shown it to have limited or variable effects [16,30,31]. The bacteriostatic or bactericidal effect of organic acid may depend on its manner of use (concentration, vehicle, and duration) and is also related to the status of the host. In our study, we found that the supplementation of malic acid in the drinking water for chicken flocks at slaughter age was effective in two chicken lines. The *Campylobacter* contamination load was decreased by 1.05–1.55 log_10_ CFU/g in the cloaca and 1.56 log_10_ CFU/g in the caeca. However, in a long period of daily treatment for three weeks during rearing, the effect of malic acid did not persist over time and significant decontamination was observed only in the first week of application. This may be caused by the tolerance mechanisms developed by *Campylobacter* or the self-adjustment of the chicken intestinal tract to decrease the effect of malic acid. Our present study suggests that the application of acidified drinking water to broilers at slaughter age could effectively reduce the infection load of *Campylobacter*. The application of malic acid before slaughter has the advantage of cost efficiency. In addition, the *Campylobacter* infection of broilers at slaughter age is epidemiologically more relevant to human infection and thus renders decontamination at this stage an important control point for *Campylobacter*. Nevertheless, the reason why a long period of application of acidified drinking water has a limited decontamination effect on *Campylobacter* still requires further investigation.

Malic acid is a flavoring agent and also an intermediate in metabolic cycles. Adding malic acid to animal diets could improve the performance of animals and the quality of their products [11,12,13]. These results showed the potential benefits of feeding malic acid to animals when suitable nutritional and managerial measures are applied. In our study, the broiler body weight, weight gain, and intestinal indices including length, weight, pH, and microbiota were not influenced by the malic acid supplementation. A previous study showed that supplementation with malic acid could influence the composition of milk and dorsal fat thickness in livestock [12,13] and, thus, its potential effect on chicken meat was analyzed. Our results showed that the protein content of the chicken meat was not influenced by the malic acid treatment, which suggests that the nutrition was maintained. Compared to the control group, the moisture was increased by 5.12% and 5.92% in thigh and breast meat, respectively, while the fat content was decreased by 1.60% in thigh meat. An increase in moisture contributes to the tenderness and juiciness of the meat [32], while a low fat content is beneficial for the consumers [33,34]; these are potential advantages and could improve the meat quality.

This study showed that malic acid-supplemented drinking water could reduce *Campylobacter* contamination in flocks. Our study only applied malic acid during the chicken rearing period and many improvements could be considered in subsequent research, such as the use of malic acid in combination with prebiotics or bacteriophages [35] as well as with nutritional, managerial, and biosecurity measures [36]. Additionally, the bactericidal effect of malic acid was observed to not only be restricted to *Campylobacter* but was also present in other food-borne pathogens [9], which indicates its promise and potential for improving the safety of poultry production.

## 5. Conclusions

This study added malic acid to the drinking water of chicken flocks to evaluate its antimicrobial effect on *Campylobacter*. A significant reduction in *Campylobacter* contamination loads was observed in the first week of supplementation and decreased over more extended periods of time. Thus, providing acidified drinking water to flocks at slaughter age for five days was shown to be an effective decontamination method. Malic acid has no adverse effects on chicken body weight, weight gain, intestinal indices, or microbiota. Meanwhile, it could change the composition of chicken meat by increasing the level of moisture and decreasing the fat content. Our results suggest that the application of malic acid to chicken flocks at slaughter age is a feasible and effective means to control the contamination of *Campylobacter*.

## Figures and Tables

**Figure 1 animals-11-01999-f001:**
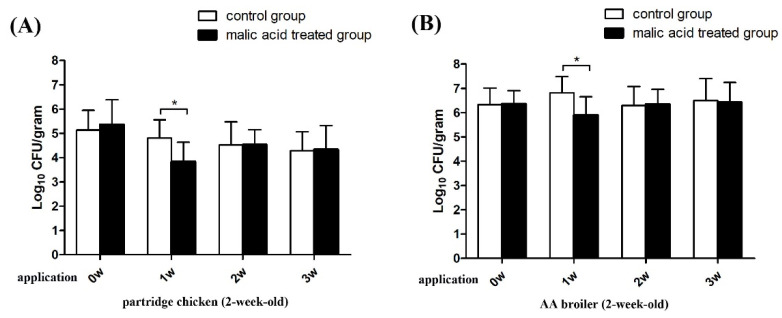
*Campylobacter* carriage in the cloaca of chickens after a long period of use of malic acid. Two week old partridge chickens (**A**) and AA broilers (**B**) were provided with the malic acid-supplemented drinking water for 3 weeks and cloacal swab samples were collected at the designated time to determine the *Campylobacter* carriage. Twenty chickens were assayed in each group, the data are presented as the mean ± SD, and asterisks indicate significant differences (*p* < 0.05 *).

**Figure 2 animals-11-01999-f002:**
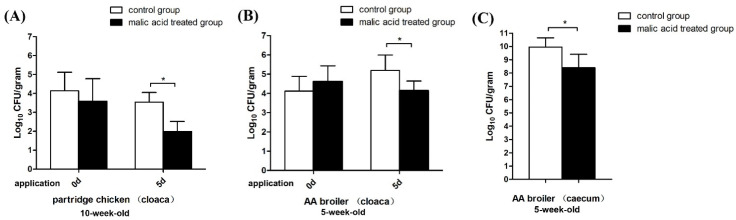
*Campylobacter* carriage in the chicken cloaca and caecum after the use of malic acid for five days. The partridge chickens (**A**) and AA broilers (**B**) at slaughter age were provided with malic acid-supplemented drinking water and the cloacal swab samples were collected before the experiment and 5 days after to determine the *Campylobacter* carriage. The caeca of the AA broilers were collected during slaughter to determine the load of colonized *Campylobacter* (**C**). Twenty chickens were assayed in each group, the data are presented as the mean ± SD, and asterisks indicate significant differences (*p* < 0.05 *).

**Table 1 animals-11-01999-t001:** Body weight (g), body weight gain (g/day), and intestinal microbiota (log_10_ CFU/g) of broilers and partridges receiving malic acid-supplemented drinking water at slaughter age.

Parameter	Broiler (5 Week Old)		*p*-Value	Partridge (10 Week Old)		*p*-Value
	Control	Malic Acid		Control	Malic Acid	
Body weight ^1^						
application of 0 d	1163.8 ± 103.8	1204.4 ± 97.6	0.39	675.2 ± 97.3	644.8 ± 53.7	0.39
application of 5 d	1504.1 ± 112.3	1523.8 ± 98.7	0.69	774.6 ± 80.4	813.3 ± 94.2	0.30
Body weight gain ^1^	76.6 ± 9.2	72.6 ± 7.6	0.36	20.1 ± 4.3	25.3 ± 6.6	0.39
Microbiota ^1^						
application of 0 d	11.67 ± 0.74	11.66 ± 0.71	0.97	11.88 ± 1.10	11.85 ± 0.54	0.95
application of 5 d	11.51 ± 0.52	11.79 ± 0.49	0.22	11.72 ± 0.93	11.66 ± 0.82	0.89

^1^ Values are given as means ± SD from 20 chickens per group.

**Table 2 animals-11-01999-t002:** Intestinal weight, length, and pH of broilers receiving malic acid-supplemented drinking water at slaughter age.

Parameter	Broiler		*p*-Value
	Control	Malic Acid	
Intestine weight (g) ^1^	25.1 ± 2.2	26.7 ± 3.0	0.16
Length (cm) of ^1^			
Small intestine	112.6 ± 1.8	110.9 ± 1.9	0.07
Caecum	11.0 ± 1.1	12.1 ± 1.2	0.06
Intestinal pH ^1^	7.26 ± 0.36	7.03 ± 0.40	0.17

^1^ Values are given as means ± SD from 20 chickens per group.

**Table 3 animals-11-01999-t003:** Chemical composition (%) of breast and thigh meats from broilers receiving malic acid-supplemented drinking water at slaughter age.

Parameter	Breast Meat		*p*-Value	Thigh Meat		*p*-Value
	Control	Malic Acid		Control	Malic Acid	
Moisture ^1^	63.25 ± 3.94	69.17 ± 2.30 *	0.02	65.66 ± 2.50	70.78 ± 2.21 *	0.01
Crude protein ^1^	21.69 ± 1.72	20.70 ± 4.96	0.68	22.91 ± 6.34	21.15 ± 2.14	0.56
Crude ash ^1^	1.15 ± 0.47	1.01 ± 0.09	0.48	1.18 ± 0.04	1.12 ± 0.20	0.48
Crude fat ^1^	3.07 ± 2.30	3.36 ± 1.76	0.85	5.59 ± 0.93	3.99 ± 0.26 *	0.01

^1^ Values are given as means ± SD from 20 chickens per group. * Asterisks indicate significant differences (*p* < 0.05) were found between the malic acid-treated group and the control group.

## Data Availability

The data presented in this study are available upon request from the corresponding author.

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
