# Peer review of "Feeding Malic Acid to Chickens at Slaughter Age Improves Microbial Safety with Regard to Campylobacter"

_animals, 2021, doi:10.3390/ani11071999_

Round 1

Reviewer 1 Report

I believe that the current version of the manuscript animals-1283947 is appropriate for publication as a communication.

Reviewer 2 Report

Thank you for reviewing and covering my comments

This manuscript is a resubmission of an earlier submission. The following is a list of the peer review reports and author responses from that submission.

Round 1

Reviewer 1 Report

See document attached

Reviewer 2 Report

The authors present a set of studies investigating the reduction of Campylobacter spp. in malic acid administered poultry. Campylobacter jejuni and Campylobacter coli are two foodborne pathogens of interest, which can be commonly isolated from broiler chickens. It is typically considered one of the more severe poultry associated foodborne pathogens. Thus, it is of great interest to reduce the possibilities of foodborne infection by decreasing the natural abundance of Campylobacter found in chickens. With regulatory concerns surrounding the use of antibiotics, and their attribution to antibiotic resistant pathogens, alternatives are extremely desired. Thus, a generally recognized as safe organic acid, like malic acid may be a logical alternative due to the antibacterial effects it has on Gram negative bacteria.

Major Comments:
Firstly, I want to acknowledge that the authors are exploring a relevant and important topic in regards to poultry and food science. The natural Campylobacter infection experiments are pertinent to an industry setting. Additionally, I find the statistics used in the manuscript to be adequate. Furthermore, I find the conclusion to be relatively sound. However, I believe the experimental design is of major concern and needs to be addressed.

Experimental Design:
Line 105-107: What does the author mean by “less individual differences”? All Male or all Female should be selected for broiler trials. Broiler chick weight should be standardized and placement should be randomized. Commercial vaccinations should be listed, if any. Furthermore, the explanation of rearing areas needs to be listed. Are these floor pens? Poultry brooders, Colony cages? Depending on the rearing method, pen effect can occur, which would make environmental conditions not uniform between treatment groups.

Line 120-127: Experiment 2 presentation is slightly confusing. Are both AA broilers and partridge chickens separated, and they total 40 chickens each, or is it 20 each? Were the AA broilers the only birds sent to slaughter? If no, were they the only birds that had intestine and meat samples collected? If so, why were the partridge chickens not subjected to the same criteria? Additionally, more confusion arises when referencing the colonization assay in a previous study. Did the authors orally inoculate the chickens like the previous paper? Or, are authors just describing the organ harvesting methods, subsequent serial dilutions, and the use of selective media?

Line 129-134:  Although malic acid may aid in disease resistance. I find Experiment 3 to be inappropriate and lacking sufficient detail. Experiment 3 should either be excluded from the manuscript or redone. Chickens suffering from an unknown respiratory disease is not an adequate disease model. Particularly when claiming it reduced mortality in chickens suffering from a respiratory disease.  Authors would need to develop a disease model with a proper bacterial or viral strain. Authors would then need to research and choose an applicable Multiplicity of Infection (MOI) in CFU or PFU. Furthermore, some of the selected poultry for experiment 3 may be further along in the disease, while others may have been in earlier stages. Which may influence the efficacy of malic acid supplementation.

Line 255-261: Authors weigh and measure the length of small intestine. What is the purpose of this? Typically, villus height, crypt depth, and their ratio are reported as indicators of intestinal health. Not just intestinal weights. What does the intestinal weights tell the researcher? Why is this pertinent? In the Discussion, it lists this as a reason why broiler performance was not impacted. Although this may be true, I would argue intestinal weights do not demonstrate that. Please address this in more detail.

Minor Comments:
Line 66-67: Author references malic acid as being Generally Recognized as Safe. I would cite the regulatory authority that deems it as GRAS.

Line 73-74: Author falsely claims that “antimicrobial efficacy during chicken rearing period has never been investigated”. Although there are only a few papers regarding malic acid, this is not the first. I would suggest removing this verbiage or citing previous papers like Moharrery, A., and M. Mahzonieh. "Effect of malic acid on visceral characteristics and coliform counts in small intestine in the broiler and layer chickens." Int. J. Poult. Sci 4.10 (2005): 761-764.

Line 78: The first appearance of “AA broilers”. What is AA abbreviated for? Arbor Acres?

Line 95: Nutrition is a driving force in the poultry industry, although the authors provide the feed composition in the supplementary table, I would reconsider adding it to the main body. Particularly if the same formulation was used for each experiment and breed.

Line 104-107: Experimental Bird selection - Why the inconsistencies in bird age? If they are broilers, why not select day old chicks and observe throughout the 42 day rearing process?

Line 139, 146, 148, 155: Microflora - this is an antiquated term. The appropriate term is microbiota.  That is because the word flora refers to plant life, which is not applicable to bacteria.

Line 152: Why did the author select CCDA agar, which is use to isolate Campylobacter from food, rather than Campy-Cefex Agar, which is designed specifically for cultivating Campylobacter species from poultry?

Line 198: Authors state that the malic acid supplement was not consistent and decreased with time. Why was the malic acid supplementation decreased?

Line 340-341: Author discusses the effectiveness of organic acids, and mentions their limited and variable effect. What specifically are those?

Line 374-378: Author demonstrates that the protein content of chicken meat was not influenced by the malic acid. Was the chicken meat subjected to a taste test? No sensory testing was added. No mouthfeel or Umami consumer results were indicated. Although the moisture and fat content may appear the same or better. The structure of the meat could be impacted by the malic acid, which could change the texture/mouthfeel of the meat. If this occurs, this would result in consumers not wanting to purchase the chicken in the future.  May want to add a food sensory survey or reference a study that determines it has no impact to consumer expectations. 

Reviewer 3 Report

Feeding Malic Acid to Chickens at Slaughter Age Benefits Poultry Production and Microbial Safety in Regard to Campylobacter

Authors: Fangzhe Ren et al.

The objective of this study was to assess the effect of feeding malic acid to chickens on Campylobacter decontamination.  Malic acid decreased the contamination of Campylobacter when administered for 5 days prior to slaughter, and had no adverse effects on chicken performance, including body weight, intestinal indices, and microflora.  However, the quality of chicken meat was improved.  Furthermore, malic reduced mortality of chickens suffering from respiratory disease.  This is an interesting article, English grammar needs to be checked.  Few other comments are below.

L 95-96 Normal water, or acidified water prepared by supplementation of L-malic acid to a final pH value of approximate 4.0, were providing to the flocks for drinking.

L 104 Please indicate if Malic Acid source and concentration used in all 3 experiments were the same.

L 106-107 Please confirm the concentration of malic acid administered

L 347-348 However, in a long period daily rearing for three weeks, the effect of malic acid was not consistent, with significant decontamination only found in the first week of application. 

Please provide a hypothesis of this effect for a long period of administration.

Round 2

Reviewer 1 Report

See document attached.

Reviewer 2 Report

I think your edits are adequate in the latest revised manuscript. Additionally, I hope your future research allows for larger bird samples sizes. Other than that, I look forward to reading about your future research.